# Behavioral Interactions between Bacterivorous Nematodes and Predatory Bacteria in a Synthetic Community

**DOI:** 10.3390/microorganisms9071362

**Published:** 2021-06-23

**Authors:** Nicola Mayrhofer, Gregory J. Velicer, Kaitlin A. Schaal, Marie Vasse

**Affiliations:** 1Institute of Integrative Biology, ETH Zürich, Universitätstrasse 16, 8092 Zürich, Switzerland; nicola.mayrhofer@usys.ethz.ch (N.M.); gregory.velicer@env.ethz.ch (G.J.V.); 2MIVEGEC (UMR 5290 CNRS, IRD, UM), CNRS, 34394 Montpellier, France

**Keywords:** microbial food web, trophic interactions, predator–prey interactions, mesopredator, social bacteria, nematodes, experimental community, behavior

## Abstract

Theory and empirical studies in metazoans predict that apex predators should shape the behavior and ecology of mesopredators and prey at lower trophic levels. Despite the ecological importance of microbial communities, few studies of predatory microbes examine such behavioral res-ponses and the multiplicity of trophic interactions. Here, we sought to assemble a three-level microbial food chain and to test for behavioral interactions between the predatory nematode *Caenorhabditis elegans* and the predatory social bacterium *Myxococcus xanthus* when cultured together with two basal prey bacteria that both predators can eat—*Escherichia coli* and *Flavobacterium johnsoniae*. We found that >90% of *C. elegans* worms failed to interact with *M. xanthus* even when it was the only potential prey species available, whereas most worms were attracted to pure patches of *E. coli* and *F. johnsoniae.* In addition, *M. xanthus* altered nematode predatory behavior on basal prey, repelling *C. elegans* from two-species patches that would be attractive without *M. xanthus*, an effect similar to that of *C. elegans* pathogens. The nematode also influenced the behavior of the bacterial predator: *M. xanthus* increased its predatory swarming rate in response to *C. elegans* in a manner dependent both on basal-prey identity and on worm density. Our results suggest that *M. xanthus* is an unattractive prey for some soil nematodes and is actively avoided when other prey are available. Most broadly, we found that nematode and bacterial predators mutually influence one another’s predatory behavior, with likely consequences for coevolution within complex microbial food webs.

## 1. Introduction

Predation is an ancient biological interaction that influences ecosystem resource turnover [1,2] as well as species abundance, diversity, and evolution [3,4,5,6,7,8]. Predators can be found at all biological scales and include organisms as different as white sharks and microbes. While less familiar, small predators such as protists, nematode worms, and bacteria make fundamental contributions to global biogeochemical cycling [9,10] and are proposed to be key players for both agriculture [11] and human health [12,13]. In addition, predation in microbial communities may have been a driving force in some of the major transitions in evolution, including the origin of the eukaryotic cell [14,15,16,17] and the advent of multicellularity [18,19].

The predatory interactions that link members of a community can be represented as food webs, trophic networks that display the flow of energy among community members. Food webs have long been a central concept in ecology and are powerful tools for investigating community structure, the nature and strength of pairwise interactions, and the indirect effects of interactions on various aspects of community ecology. Food web research typically relies on direct observation of organism behavior, but such direct observations are difficult or impossible when studying microbes (especially given that many microbes cannot be cultured under laboratory conditions). To study species’ interactions within microbial food webs [20], ecologists rely on computational studies, mathematical modelling, and experiments in simplified microbial systems. Examples include flux-balance analysis [21,22,23], study of pairwise interactions, and examination of growth and death curves in small communities [24]. These approaches usually adopt a bottom-up strategy, inferring features of the community based on the careful investigation of its components. However, they likely miss out on higher-order interactions and complex behavioral responses. Since predatory interactions may occur in complex networks, involve various partners, and fluctuate over time, these bottom-up approaches might be insufficient to understand the dynamics and broader ecological impacts of microbial predation [25,26,27].

Despite the ecological importance of microbes [10], microbial predators have only recently received substantial recognition as agents that influence biodiversity by controlling and shaping bacterial communities [9,28,29,30,31,32]. Microbial predators use a wide range of strategies to kill and consume their prey, from the periplasm-invasion strategy of *Bdellovibrio bacteriovorus*, which grows and divides within its prey, to total engulfment by protists and far-range killing by *Streptomyces* species [33]. Cells of *Myxococcus xanthus*, the most studied myxobacteria species, forage in groups, repeatedly reversing direction while attacking and consuming prey [34]. Myxobacteria are predicted to play major roles in shaping the structure and evolution of soil communities due to their ability to eat diverse species as prey [35] and strongly influence prey evolution [36], as well as their abundance in soils [37].

According to current understanding, *M. xanthus* secretes extracellular hydrolytic enzymes that break down prey macromolecules and allow uptake of the released nutrients [38]. Studies of *M. xanthus* predation have examined its molecular mechanisms [34,39,40], the effects of ecological conditions [35,41,42,43], and (co)evolution with a single prey species [36,44]. Little is understood about how *M. xanthus* may itself be exposed to predation pressure and how it interacts with its own predators. Natural bacterial communities are often grazed by bacteriophagous microfauna such as nematodes and protozoa, which can influence their structure and composition [11,45]. It is very likely that some such bacte-riophagous organisms prey upon *M. xanthus* in natural environments, thus making *M. xanthus* a potential mesopredator, defined as an organism in a given food web that obtains nutrients by killing and consuming other organisms and faces predation risk from larger organisms [46]. In fact, Dahl et al. [47] showed that the predatory nematode *Caenorhabditis elegans* will ingest *M. xanthus* in some contexts. However, whether *C. elegans* achieves net growth from nutrients derived from wild-type *M. xanthus* remains uncertain.

It is not known to what extent the community-ecology effects of microbial predators mirror those of multicellular predators (but see Steffan et al. [48]). In large organisms, intraguild predation (when apex predator and mesopredator also compete for the same basal prey organism) can have direct effects on mesopredator survival and distribution [49]. Ritchie and Johnson [50] reviewed the effects of apex predators on mesopredators and their prey in 94 animal studies and found that, on average, increasing apex predator population size two-fold reduces mesopredator abundance by approximately four-fold. Such effects may be similarly important in communities of microbes. In addition to their direct demographic effects, apex predators can generate substantial behavioral modifications in mesopredators, altering their habitat use and changing their foraging activity, thereby indirectly affecting their survival and growth [50]. It is unclear whether these communities show hierarchical trophic interactions, and, if so, whether bacteriophagous organisms function as apex predators. The study of microbial community dynamics, therefore, requires a better understanding of the direct and indirect interactions between bacterial prey, bacterial predators (potential mesopredators), and bacteriophagous nematodes and protozoa (potential apex predators).

Here, we examined behavioral interactions between two predator species in a synthetic community and investigated how such interactions modulate food web structure. We designed the community to have three trophic levels. Predicted trophic interactions are depicted in Figure 1. We hypothesized *C. elegans* to function as a potential apex predator, *M. xanthus* as a potential mesopredator, and *Escherichia coli* and *Flavobacterium johnsoniae* as two basal prey bacteria. We first tested for effects of the bacterial predator on nematode predatory behavior by asking whether (1) *M. xanthus* attracts or repels *C. elegans* in the absence of other prey, (2) bacterial cell death or strain motility alters any effect of *M. xanthus* on *C. elegans,* (3) *M. xanthus* is more or less attractive to *C. elegans* as potential prey than the two basal prey species, and (4) the presence of *M. xanthus* in mixture with one basal prey species in a given prey patch alters its attractiveness to worms. We then asked whether nematodes reciprocally influence *M. xanthus* behavior—specifically, swarming behavior within patches of basal prey—whether due to direct interactions between the predator species or indirectly due to nematode effects on basal prey populations.

## 2. Materials and Methods

### 2.1. Bacterial and Nematode Strains

As the hypothesized apex predator, we used *C. elegans* strain N2 (CGC). As hypo-thesized mesopredators, we used two strains of *Myxococcus xanthus,* GJV1 and GJV71. *M. xanthus* uses two distinct motility systems to drive swarming across solid surfaces, traditionally referred to as the ‘A motility system’ and the ‘S motility system’ [51]. Strain GJV1 possesses both systems functionally intact and is a clone of DK1622 [52], which was used in the only prior study reporting interactions between *M. xanthus* and *C. elegans* [47]. For purposes of this paper, we hereafter refer to GJV1 as strain S, for ‘swarming’. GJV71 is a nonmotile mutant of GJV1 with major deletions in two genes, one gene essential for A motility (*cglB*) and one gene essential for S motility (*pilA*). GJV71 was referred to as strain ‘A1 *cglB*’ in Velicer and Yu [53]. We hereafter refer to GJV71 as strain N, for ‘non-swar-ming’. We selected the *Escherichia coli* strain OP50 [54] (CGC, Caenorhabditis Genetic Center) and *Flavobacterium johnsoniae* (ATCC^®^ 17061™) as basal prey bacteria because they represent, respectively, high- and intermediate-quality food sources for *M. xanthus*, promoting *M. xanthus* swarming and growth to different degrees [35,43]. *E. coli* strain OP50 is the standard prey for laboratory populations of *C. elegans* [55]. In preliminary experiments, *F. johnsoniae* sometimes displayed a phenotype with low gliding ability, which inhibited *M. xanthus* predation. In all following experiments, the source of *F. johnsoniae* was a frozen stock originating from a single colony, which we isolated from a normally sprea-ding population.

### 2.2. Standard Culture Conditions

Unless otherwise indicated, organisms were cultured on 6-cm diameter petri dishes each with 14 mL of 1.5% agar CFcc medium (‘clone fruiting’ medium [56] supplemented with 1 mM CaCl_2_ and 0.005 mg/mL cholesterol).

### 2.3. Culturing C. elegans

We froze *C. elegans* in a 10% DMSO solution and thawed it in minimal salts buffer (M9) with glutamine, according to Pires da Silva et al. [57]. We maintained the worms at room temperature on 1.5% agar nematode growth medium (NGM) dishes seeded with *E. coli* OP50, transferring weekly. We synchronized the life stages of all *C. elegans* populations prior to use in an experiment. Seven days before the start of the experiment, we transferred a small inoculum from a growing population to seeded 1.5% agar high growth medium (HGM) dishes [58]. After 6 days of incubation at room temperature, we washed the agar surface with M9 to collect the worms. We centrifuged them at 173× *g* for 1 min and removed all but 1 mL of supernatant. We then added 3 mL of bleaching solution (6 mL ddH_2_O, 6 mL NaOCl (5% Cl), 2 mL 1M NaOH) and waited up to 6 min, vortexing every 2 min. This step dissolved the bodies of the adult worms, releasing the eggs. We then washed the released eggs four times by centrifuging, removing all but 500 µL of supernatant, and adding ddH_2_O to 5 mL. After the final wash, we added 3.5 mL of M9 and transferred the egg suspension to a 6-cm petri dish to hatch at room temperature overnight. To prevent contamination, we added 40 µg/mL gentamicin. The next day, we collected the hatched L1 larvae by centrifuging and resuspended them in CFcc liquid. We determined the worm density in the suspension by plating three 1-µL drops on unseeded CFcc agar plates and counting the worms in each drop.

### 2.4. Culturing M. xanthus

We inoculated *M. xanthus* from freezer stock onto 1.5% agar CTT (10 g/L Casitone, 10 mM Tris pH 8.0, 8 mM MgSO_4_, 1 mM KPO_4_ [59]) dishes and incubated it at 32 °C and 90% relative humidity (rH) for 4–5 days. We then sampled the outer edge of the resulting colony and transferred the inoculum into CTT liquid, shaking at 32 °C and 300 rpm for 1 day until the cultures reached mid-exponential phase, then adjusted them to an absor-bance (OD_600_) of 5 in CFcc liquid.

### 2.5. Culturing Prey Bacteria

We streaked *F. johnsoniae* and *E. coli* from freezer stock onto 1.5% agar lysogeny broth (LB, Sigma, St. Louis, MI, USA) dishes and incubated them at 32 °C and 90% rH for 3 days. We transferred single colonies into LB liquid, shaking at 32 °C and 300 rpm for 1 day (or ~10 h in the case of *E. coli*), then adjusted the cultures to an OD_600_ of 5 in CFcc liquid.

### 2.6. C. elegans’ Binary Choice Assays

We inoculated two 15-µL bacteria spots 2 cm apart on CFcc agar and incubated at 25 °C and 50% rH overnight before adding *C. elegans*. We added 20 *C. elegans* L1 larvae suspended in CFcc liquid to the dishes and incubated them at 25 °C and 50% rH. We counted the number of worms in different regions of the petri dish under a dissecting microscope at several time points. For the assay reported in Figure 2B, we observed the worms after 1 h and 18 h, as preliminary experiments indicated that the worm location did not change after 18 h. For the assays reported in Figure 3 and Figure 4, we used the same time points plus an additional one at 25 h to allow the prey and *M. xanthus* to interact and this interaction to potentially affect the worm location. For the assays reported in Figure 2A,B, we prepared both living and dead cultures of *M. xanthus* strains S and N. To kill *M. xanthus*, we resuspended a growing culture to OD 5 in CFcc liquid and incubated at 50 °C for 3 h. We left live cells shaking at 32 °C and 300 rpm during this time, before adjusting the OD. An illustration of the *C. elegans’* binary choice assay design is depicted in Figure 1. Results from these assays are shown in Figure 2A,B, Figure 3 and Figure 4.

### 2.7. C. elegans Half-Plate Choice Assays

We drew center lines on CFcc plates and prepared both living and dead cultures of *M. xanthus* strains S and N as reported above. We inoculated one half of each petri dish with 100 µL of one of the bacterial cultures or buffer control spread with a 10-µm loop and allowed the inoculum to dry. We bleached *C. elegans* and adjusted the egg suspension to 50 eggs/µL by counting the number of eggs in three 0.5-µL drops. We immediately added 20 µL of egg suspension (approximately 1000 eggs) to each dish along the center line. We incubated the dishes at 25 °C and 50% rH and counted the number of worms on each side of the dish under a dissecting microscope at several time points. We chose the time points of 18 and 42 h to be similar to the time points for the experiment in Figure 2B but with a delay to allow the nematode eggs to hatch on the dish. The plating design of the *C. elegans* half-plate choice assays is illustrated in Figure 1. Results from these assays are shown in Figure 2C,D.

### 2.8. M. xanthus Swarming Assays

We marked CFcc plates with reference lines and scale bars for image analysis. We inoculated 15 µL each of *F. johnsoniae*, *E. coli*, and *M. xanthus* strain S in a row, with *M. xanthus* in the middle and 1-cm distance between each inoculation spot, and incubated the dishes at 20 °C and 50% rH. We prepared *C. elegans* worms by bleaching them on the same day that we plated the bacteria, and the next day we added the appropriate number of worms to each dish either by manually picking the desired number of individual L1 larvae and adding them directly or by adding 10 µL of worms suspended in CFcc liquid adjusted to the appropriate concentration. We took pictures of the experimental plates every 24 h, and we measured the distance *M. xanthus* swarmed into each prey patch over time with image analysis using Fiji [60]. Figure 1 shows an illustration of the *M. xanthus* swarming assays. Results from these assays are shown in Figure 5.

### 2.9. Statistical Analysis

We performed all data analysis and statistical testing using R version 3.6.2 and RStudio version 1.2.5033 [61,62]. We tested the effect of *C. elegans* on *M. xanthus* swarming distance on prey using a mixed linear model with prey type (*E. coli* or *F. johnsoniae*) and worm treatment (factor presence/absence in one experiment, continuous variable number of worms in another) as fixed effects. As we measured the swarming distance on the two prey species from the same experimental petri dish, we included the dish identity as a random factor to account for repeated measures. We compared treatment modalities u-sing the Tukey method for multiple comparisons from the emmeans package version 1.4.3 [63]. To evaluate *C. elegans*’ choice in the binary choice and half-plate choice assays, we calculated a choice index as in Moore et al. [64]:(# worms on side A − # worms on side B)/(# worms on side A + # worms on side B).(1)

Null values indicate that the worms did not prefer one side over the other (or that they all left the dish). To compare their attraction to or avoidance of live and dead *M. xanthus* (strain S or N), we used an ANOVA with the options treatment and time as fixed effects. We considered time as a fixed effect because we were interested in whether the differences between the option treatments would change over time. We performed post hoc comparisons with the Tukey method. We further tested whether the worms preferred one option over the others with one-sample *t*-tests against 0 with Bonferroni correction for multiple testing.

## 3. Results

### 3.1. Only a Few C. elegans Worms Interact with M. xanthus Regardless of Whether It Is Alive or Dead

To test for interactions between *M. xanthus* and *C. elegans*, we co-cultured worms and bacteria on agar petri dishes in a variety of assays. In all of these assays, the worms had three spatial areas to choose among: (1) outside the assay plate, which could be reached by worms climbing out of the petri dish; (2) agar-surface regions with no bacteria present; and (3) agar-surface areas covered by bacterial cells, with some of these areas being circular patches (binary choice assays) and others covering half of a petri dish (half-plate choice assay).

In our first experiments, only *M. xanthus* was available as potential prey, and we offered both live and dead *M. xanthus* cells to *C. elegans* to test for any effect of cell death on their attractiveness. If *M. xanthus* is an attractive prey item for *C. elegans,* it might attract worms equally whether alive or dead. Alternatively, if *C. elegans* avoids living *M. xanthus* cells because they produce a repellant compound, dead *M. xanthus* might, nonetheless, serve as a palatable food source. Two strains of *M. xanthus* were offered to *C. elegans,* one motile (strain S) and one non-motile (strain N).

In a binary choice assay, we inoculated two circular patches of *M. xanthus* on an agar surface and added L1 larval worms to a bacteria-free region of the plate, equidistant from the two *M. xanthus* patches. In this assay, we counted how many worms left the plate vs. remained on the plate after 1 and 18 h and, of those that remained, how many entered one or the other of the *M. xanthus* patches. In most replicates, regardless of the options provided, large majorities of the worm populations emigrated from the dish (>75% on average), and there was no general difference in the rate of emigration as a function of *M. xanthus* strain identity (ANOVA bacterial identity factor *F_2,12_* = 0.8, *p* = 0.5, Figure 2A). Those who stayed demonstrated no clear general preference between live vs. dead patches across both strains and both examined time points (1 and 18 h, ANOVA choice:time interaction *F_2,24_* = 5.67, *p* < 0.01, post hoc Tukey HSD tests *p* > 0.2; Figure 2B and Appendix A). One exception to the general lack of a strong effect of *M. xanthus* death occurred on dishes containing the motility mutant strain N. On these plates, the worms seemed to initially prefer the live strain N patch after 1 h but then changed their preference to the dead patch by 18 h (post hoc Tukey HSD test *p* = 0.01). Because no similar pattern was seen for strain S, this result suggested that the effect of death on the attractiveness of bacterial cells to nematode predators can vary across conspecific genotypes.

The preference of the worms to leave the experimental petri dishes suggested that *M. xanthus* might repel *C. elegans*. To test this hypothesis, we inoculated *M. xanthus* alone—either strain S or N, alive or dead—onto half of the agar surface of the petri dish, leaving the other half uninoculated. In the previous binary choice assay, the patches of *M. xanthus* were small relative to the agar surface of the experimental petri dishes, reducing the likelihood of finding worms in a patch unless the bacteria actively attracted them. In contrast, in this half-plate choice assay we expected to find 50% of the worms on the plate located in the bacterial lawn, assuming no interactions between the two organisms. We included plates inoculated with sterile resuspension buffer to control for the potential attractive or repulsive effect of the buffer itself. We added *C. elegans* eggs to the midline of the dish and counted how many worms left vs. remained on the plate after hatching and, of those that remained, how many went to each half of the dish.

The overwhelming majority of worms (>90%) preferred to leave the dish both in the absence and presence of *M. xanthus* (Figure 2C). However, a larger proportion of worms remained on the plate when *M. xanthus* was present (~4% on average across all treatments with *M. xanthus*) than on control plates with buffer alone (~1%, *t_22_* = 6.42, *p* < 0.01). Moreover, the worms that remained were more likely to be located on the inoculated side of the dish on plates with *M. xanthus* versus on plates with only buffer (ANOVA bacterial presence *F_2,19_* = 27.43, *p* < 0.01, post hoc Tukey HSD test for preference of strain S or strain N over buffer, *p*-values < 0.01; Figure 2D and Appendix A). We saw no difference between live and dead (post hoc Tukey HSD test *p* = 0.24) or motile and non-motile *M. xanthus* (post hoc Tukey HSD test *p* = 0.44). Our results revealed intrapopulation heterogeneity in whether worms remain on plates containing only *M. xanthus* and suggested that, among the minority of worms that did remain, *C. elegans* was attracted to *M. xanthus*.

### 3.2. C. elegans Prefers Both Basal Prey Species over M. xanthus

To investigate potential behavioral responses of *C. elegans* to *M. xanthus* relative to other potential prey, we performed additional binary choice assays. We inoculated two patches of bacteria on each dish, allowing the worms to choose between them, remain in the open agar, or emigrate from the dish, and then counted the number of worms in each plate area after 1, 17, and 25 h. In these experiments, each patch contained either one basal prey species alone, one basal prey mixed 1:1 with *M. xanthus* strain S or N, or *M. xanthus* strain S or N alone. We added 20 worms to a bacteria-free region of the plate, allowing them to explore across the agar surface and seek out their preferred prey.

As expected from the previous results (Figure 2A,C), when *M. xanthus* was the only option, most worms had either left the dish entirely or were located in the open agar after 25 h (Figure 3A and Appendix A). When *C. elegans* could choose between a patch of basal prey mixed with *M. xanthus* and a patch of the same prey without *M. xanthus*, the worms almost invariably chose the latter (ANOVA *F_13,28_* = 10.01 *p* < 0.01, *p*-values < 0.05 except for *F. johnsoniae* vs. *F. johnsoniae* + strain S; Figure 3B and Appendix A). In the absence of *M. xanthus*, *C. elegans* preferred *E. coli* over *F. johnsoniae* (one-sample *t*-test for choice index < 0 *t_2_* = −3.47 *p* = 0.037; Figure 3C, Figure 4, Appendix A). However, the presence of *M. xanthus* in *E. coli* patches altered the preference of the worms; *C. elegans* tended to prefer patches of *F. johnsoniae* over mixed patches containing both *E. coli* and *M. xanthus* (*p* = 0.096 and *p* = 0.028 for mixes with strain S and strain N, respectively, 14 two-sided *t*-tests with Bonferroni–Holm correction; Figure 3C, Figure 4, Appendix A). In general, the presence of *M. xanthus* drastically reduced patch attractiveness for *C. elegans* at every time point, independent of *M. xanthus* strain identity (*p* < 0.1, *t*-tests as described above; Figure 3, Figure 4, Appendix A).

Depletion of prey patches by *M. xanthus* typically requires many hours or even several days, depending on the prey type [42,43]. In this experiment, many worms localized to either Patch 1 or Patch 2 already after one hour, and by 17 h nearly all did so (among the ones remaining on the dish, Appendix A). We considered it unlikely *M. xanthus* had by that time fully cleared the prey bacteria from the mixed patches. However, to confirm that *C. elegans* did not change its behavior toward the basal prey simply as a consequence of *M. xanthus* having consumed all basal prey within the patch, leaving none to tempt *C. elegans*, we streaked samples of each mixed patch on LB petri dishes at the end of the experiment to check for the presence of *E. coli* or *F. johnsoniae*. In seven out of 24 cases, we could verify that the prey bacterium was still present in the mixed patch after 25 h (Appendix A). Even in these cases, the worms tended not to go into the mixed patches, suggesting that the presence of *M. xanthus* in mixed bacterial patches repelled *C. elegans* toward pure patches of *E. coli* or *F. johnsoniae*.

### 3.3. M. xanthus Responds Behaviorally to C. elegans in a Prey-Dependent Manner

To characterize potential behavioral responses of *M. xanthus* to the presence of *C. elegans*, we performed a binary choice assay similar to those above. In this assay, we added *C. elegans* to an agar petri dish inoculated with two patches of basal prey, one of *E. coli* (e.g., Figure 5A) and one of *F. johnsoniae*, and one patch of *M. xanthus* at the midpoint between the prey such that it would encounter them upon swarming outward. Nematodes might alter *M. xanthus* swarming behavior due to direct interactions with *M. xanthus* or due to indirect effects of resource competition for basal prey, which, in turn, might be affected by the type of basal prey environment.

We measured *M. xanthus* swarming rate in each prey-patch type on dishes with (Figure 5A) and without worms. This swarming-rate measure encompasses both the ability to penetrate the prey patch and predatory performance inside the patch [43]. For the treatment with *C. elegans*, we added 10 worms to a bacteria-free region of the plate. The nematodes had no effect on *M. xanthus* swarming rate in the *F. johnsoniae* patches (ANOVA basal prey identity:worm presence interaction *F_1,36_* = 8.82 *p* < 0.01, post hoc Tukey HSD test *p* = 0.99; Figure 5B,C) but significantly increased *M. xanthus* swarming in the *E. coli* patches (post hoc Tukey HSD test *p* = 0.0004; Figure 5B,C).

The effect of worms in the *E. coli* patches only became visible after day 5 (Figure 5C), and we hypothesized that this could be explained by the onset of worm reproduction, as the worms reached maturity, and subsequent increase in the worm population size. To evaluate whether larger populations of *C. elegans* lead to an increase in *M. xanthus* swarming rate, we repeated the experiment using different numbers of worms. For this experiment, we show the swarming distances between days 3 and 5 (rather than days 5–8 as in the previous assay) to capture the response of *M. xanthus* to the inoculated number of worms rather than to a growing population, as within this time frame the worms had not yet completed their development. We found that *C. elegans* increased *M. xanthus* swarming on *E. coli* to a similar small degree regardless of worm population size (linear model *F_1,10_* = 0.82, *p* = 0.39, adjusted R^2^ = −0.01; Figure 5D). It, therefore, remains unclear whether the time delay is simply a delay in *M. xanthus* response or whether it has to do with the developmental progress of the worms. However, in contrast to our first experiment, *C. elegans* clearly increased *M. xanthus* swarming rate on *F. johnsoniae,* but did so only as a function of worm population size (linear *F_1,10_* = 32.01, *p* < 0.001, adjusted R^2^ = 0.74; Figure 5D), such that an effect of *C. elegans* was only evident when hundreds of worms were added.

## 4. Discussion

Despite their importance in microbial population turnover and community dyna-mics, there has been little research on microbial trophic chains. Here we investigated interactions between two bacterivorous predators, *M. xanthus* and *C. elegans*, in the context of a synthetic community that included two species of basal prey (*E. coli* and *F. johnsoniae*). We found that *M. xanthus* generally repels *C. elegans* relative to the effects of the two basal prey. When *M. xanthus* was the only prey option available, most worms departed our experimental predation dishes (Figure 2A,C and Figure 3A), whereas when either or both of the basal prey were offered most of the worms remained on the plates (Figure 3B,C). *M. xanthus* was not entirely repulsive to worms though. When only *M. xanthus* was offered, the few worms that remained on the plate localized more frequently within areas with *M. xanthus* than on open agar (Figure 2D). However, the presence of *M. xanthus* in a prey patch mixed with a basal prey species repelled *C. elegans* when a separate monoculture patch of either of the basal prey species was available (Figure 3B,C and Figure 4). We further showed that *C. elegans* can alter *M. xanthus* behavior by increasing the bacterial predator’s swarming rate across patches of basal prey, and that such behavior alteration depends on basal-prey identity. Our results highlight the importance of predator–predator interactions in microbial communities and the idea that other community members may often considerably modify pairwise behavioral interactions between organisms in such communities.

Theory and previous experiments in metazoans predict that the behavioral interactions between apex predators and mesopredators are driven mainly by the former [65,66,67]. In our microbial system, however, when a pure patch of basal prey was available, *C. elegans* avoided foraging areas occupied by *M. xanthus* even when they also contained the worms’ preferred prey (Figure 3 and Figure 4). One limitation of our study is that we assessed the behavior of the worms based on discrete time points instead of continuous observation. However, the worms’ choice of basal prey patches did not change between three different time points, suggesting that most worms remained in the prey patch they had initially entered (Appendix A). Although in this study we did not formally test whether *C. elegans* can use *M. xanthus* as a food source to fuel worm population growth, our results suggest that the behavioral interactions between the two may prevent *C. elegans* from ac-ting as an apex predator in this system. Despite Dahl and colleagues’ conclusion that *C. elegans* can be a predator of *M. xanthus* [47], we found that *C. elegans* worms seem to consider *M. xanthus* to be an unpalatable food source. They avoid it whenever more palatable prey is available (Figure 3B,C and Figure 4), and a majority of individuals avoid it even when there is no other prey available (Figure 2A,C and Figure 3A); *C. elegans* does not similarly avoid the basal prey (Figure 3B,C). Still, minorities of worms did interact with *M. xanthus* (Figure 2A,C and Figure 3A) and preferred it over sterile buffer (Figure 2C,D). This could reflect a level of behavioral heterogeneity in the worm population, potentially due to different feeding preferences, predatory behaviors, or sensitivity to repellant compounds produced by *M. xanthus* across individual worms. Such behavior by some worms indicates that there may be some conditions under which *C. elegans* can be attracted to *M. xanthus,* for example, when no other prey source is available.

*C. elegans* uses its nervous system to recognize different bacteria in its environment [68] and to modify its locomotive behavior in response to prey quality [69]. It can learn to recognize and approach high-quality prey [69] and to avoid pathogens [70]. There is evidence that learned pathogen avoidance is modulated by changes in gene expression that are heritable through four generations [64]. While the mechanistic reasons for the avoidance behaviors we observed here remain to be investigated, these behaviors are consistent with the hypotheses that (i) *M. xanthus*’ large secondary metabolome [39,71,72,73] contains some compounds with a primary or secondary defensive function against predators [74], and (ii) the worms can either sense them at a distance or learn to avoid them after the first encounter [70]. For example, *C. elegans* is known to avoid some *Serratia marcescens* strains after coming into contact with the serrawettin surfactants that the bacteria use for swarming motility [75]. As *M. xanthus* A-motility in particular involves the secretion of a polysaccharide surfactant [76], a similar avoidance mechanism may be involved here in the interactions between worms and the live *M. xanthus* S strain. The same compounds may also modulate interactions with the live N strain, as the mutation which knocks out A-motility may not affect production of the relevant compounds. The extent to which such compounds are repellant for the worms may be modulated by the basal prey species, as distinct metabolites may be produced in the context of different multispecies setups [77]. *M. xanthus* may produce them only during predation, or they may be repellant only compared to the more attractive compounds produced by the basal prey. If such compounds are discovered, it would be of interest to investigate whether they have specific targets or can repel a broad range of predators, to assess the importance of chemical warfare in *M. xanthus*’ trophic interactions.

Our results, however, do not support the hypothesis that *M. xanthus* facultatively produces repellant compounds in response to the presence of *C. elegans*, as dead bacteria (that no longer produce any compounds) were not collectively less repellant to the worms than live bacteria (Figure 2). Alternatively, we can hypothesize that repellant compounds produced constitutively or with a different original purpose during the growth phase of *M. xanthus* remain active, at least partially, in the inoculum and deter *C. elegans* from interacting with the bacterial predator even after it is dead. For strain N in the binary choice assay, the worms that stayed on the plate were more likely to interact with the dead bacteria after 18 h than after only 1 h (Figure 2B). It is possible that after 18 h the repellant compounds had been diluted or degraded to a level that allowed *C. elegans* to interact with the dead cells. We might expect this effect to be observed more noticeably in strain N than in strain S because strain N’s inability to swarm might result in a higher concentration of any secreted compounds in the vicinity of the living bacterial colony, creating a stronger contrast between the live and dead treatments of strain N than of strain S. On plates with live *M. xanthus*, we observed that the worms which entered areas with *M. xanthus* tended to aggregate around fruiting bodies. Cells undergoing development would likely decrease active production of repellant compounds in order to devote cellular resources to the developmental process. These hypotheses merit further investigation.

The impact of apex predators on mesopredators goes beyond killing effects to include indirect behavioral changes. Some predator-induced behavior changes do not require direct contact between the predator and its prey. Animal mesopredators commonly observe traces of an apex predator (e.g., scat) and modify their foraging strategies to avoid certain areas or times of day in order to reduce their own predation risk [50,78,79,80]. Such non-lethal effects, often called risk effects, shape not only mesopredators’ behavior but also their reproduction and survival [81,82], with cascading impacts on ecosystem structure [83]. In our model system, the presence of *C. elegans* in the arena modified the predatory behavior of *M. xanthus*, even though the worms rarely interacted directly with the *M. xanthus* swarm. This effect was modulated by the basal prey identity, suggesting that prey species may exert a potential bottom-up control on the interactions between predators [84,85,86]. When the basal prey was *F. johnsoniae, M. xanthus’* swarming rate on the prey patch depended on the density of worms. In contrast, *M. xanthus* swarmed faster on *E. coli* in the presence of *C. elegans* irrespective of worm density (after an initial delay in the response; Figure 5). It is possible that lower attraction of the worms to *F. johnsoniae* explains the density-dependent response in *M. xanthus*: when the density of *C. elegans* is low, there are often, by chance, only very few worms in the vicinity of *M. xanthus* when it preys on *F. johnsoniae* as opposed to when it preys on *E. coli*, which is statistically less likely as the worm density increases. Such differential effect of *C. elegans* on the interaction between *M. xanthus* and the basal prey could further result from the worms carrying cells of the bacterial predator to new locations as they move around the dish. However, we would expect to see this represented in the growth pattern of *M. xanthus* on the plate by the end of the experiment, for example, as tendrils of growth emanating away from the main bacterial colony. Even after 8 days, we still saw a very distinct edge of the *M. xanthus* swarm and no visible growth outside of the main patch. We, therefore, expect the *M. xanthus* cells that could have been moved to new locations by worms to play a negligible role in the swarming rate of the *M. xanthus* patch.

The Mesopredator Release Hypothesis (MRH, e.g., [66,87]) predicts that interference interactions between apex predators and mesopredators can have profound effects on regional ecosystem structures and large-scale biomass distribution patterns. These effects have been observed in studies of animal communities [65,88]. According to the MRH, reduction in an apex predator population liberates mesopredators both from killing effects and from the need for risk-reduction behaviors. As a result, mesopredator populations increase and individuals forage more freely, which can decimate prey populations. Interference effects between top predators and mesopredators should, therefore, be considered in order to understand how ecosystems are shaped. Johnke and colleagues [25] showed that, in microbes, the combination of generalist, semi-specialist, and specialist predators can help maintain overall diversity and prevent extinction of prey species due to interference competition among the predators. However, these predators did not ne-cessarily prey on each other, and it is not known how the addition of such effects may have altered the outcome. Such studies provide valuable insight into factors maintaining diversity in microbial communities, but they are unable to address questions about more complex trophic dynamics and, in particular, the ways in which apex predators may control populations of microbial mesopredators.

In metazoans, behavioral observation often constitutes a key source of information about indirect interactions that, as previously mentioned, can alter both food web structure and dynamics, sometimes more strongly than density-mediated effects (reviewed by Werner and Peacor [89]). Our results do not yet provide a clear picture of the factors go-verning interactions between *C. elegans* and *M. xanthus*, but they offer a starting point for developing model experimental systems that allow systematic behavioral observation in nematodes and bacteria. Given the crucial role of organism behavior in the structuring of metazoan food webs, we emphasize the need for microbial food web studies to investigate behavior-mediated effects as well as direct killing effects. We suggest that *C. elegans* may not be an ideal candidate for the role of apex predator, given unclarity regarding its ability to prey on *M. xanthus*, but perhaps a protozoan or another nematode such as *Pristionchus pacificus* would more readily feed on *M. xanthus*.

In microbial communities, the overlap between ecological and evolutionary time scales has generated a number of insightful studies on food web dynamics and between-predator interactions [88,90,91,92,93]. However, most work has, to date, focused on density-mediated effects of interactions, and conceptual strategies for studying behaviors of predators of microbes remain scarce. Our synthetic community constitutes one step forward in that direction.

## Figures and Tables

**Figure 1 microorganisms-09-01362-f001:**
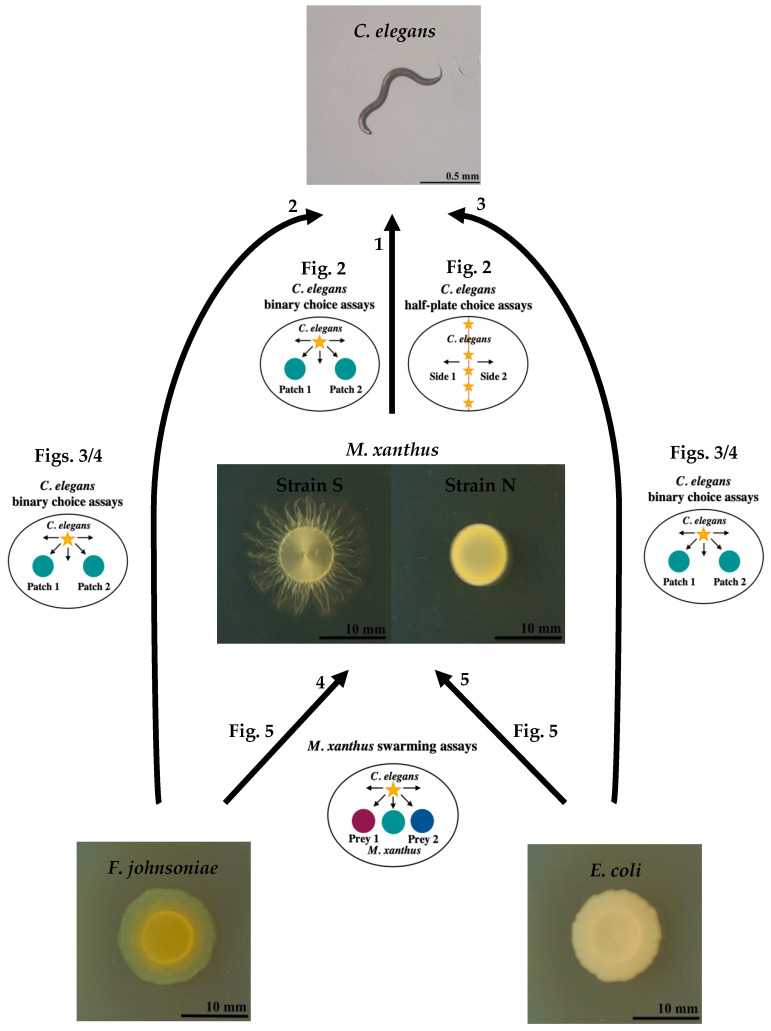
**Predicted trophic interactions of the synthetic community and illustrations of experimental designs.** We predicted that *C. elegans* might function as a potential apex predator in this food web, preying on all other members of the synthetic community (arrows 1, 2, and 3). We predicted *M. xanthus* to function as a mesopredator that preys upon both basal prey species (arrows 4 and 5) and to experience predation pressure from the nematode apex predator (arrow 1). We show illustrations of the experimental designs that were used to test each depicted interaction and reference the figures that report the relevant results.

**Figure 2 microorganisms-09-01362-f002:**
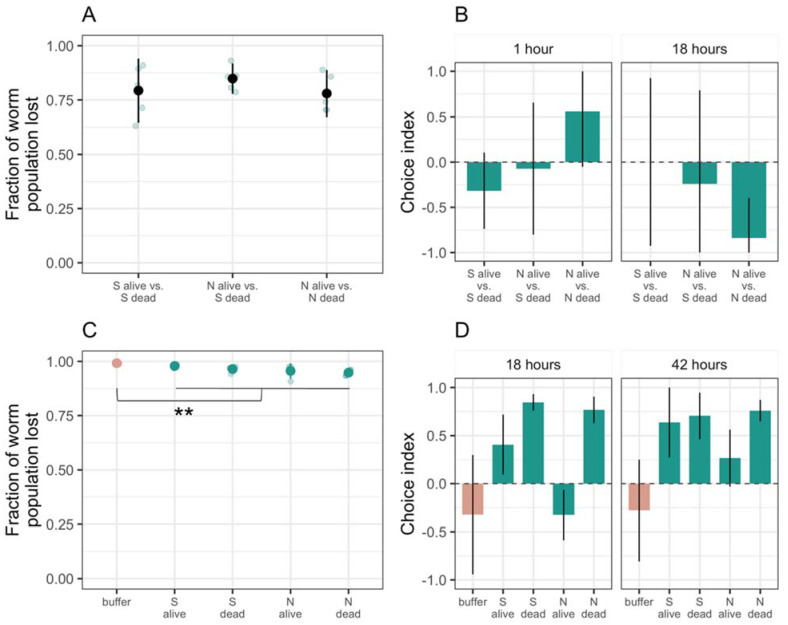
***C. elegans* prefers *M. xanthus* to buffered agar.** We tested potential effects of living or dead *M. xanthus* on the position of worms presented with choices between either two circular patches of *M. xanthus* (**A**,**B**) or between a half-plate lawn of *M. xanthus* versus a half plate of bacteria-free agar (**C**,**D**). For both experiments, we report the mean fraction of *C. elegans* populations that left the plate (**A**,**C**) and the choices made by the worms that remained on the plate (**B**,**D**). In panel B, positive vs. negative values reflect attraction to the first- vs. second-listed type of bacteria, respectively. In panel D, positive vs. negative values reflect attraction to the inoculated vs. uninoculated half of the plate, respectively. Each large dot is the mean of five biological replicates (shown as transparent blue dots). Error bars represent 95% confidence intervals. In some instances (**B**), 95% confidence intervals extend outside the range of what is biologically possible, [−1,1], so we restricted them to reflect the biological reality. ** *p* < 0.01.

**Figure 3 microorganisms-09-01362-f003:**
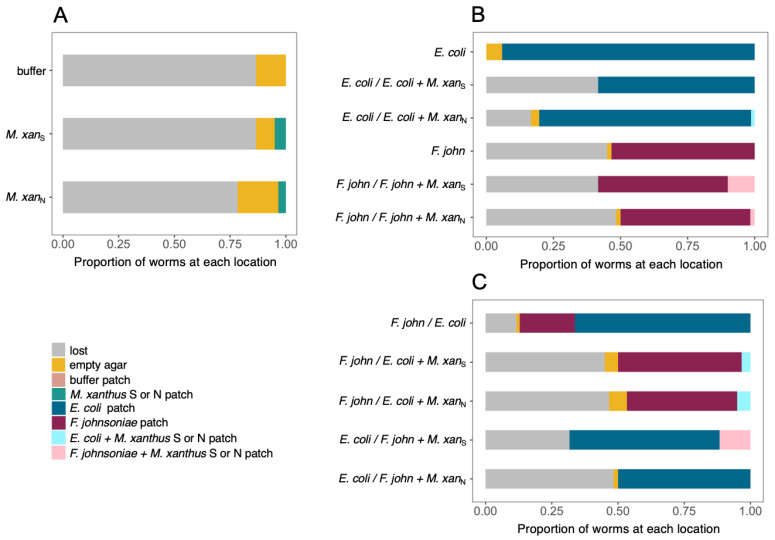
***C. elegans* prefers prey patches not containing *M. xanthus*.** Here we show worm locations relative to two circular prey patches after 25 h on plates where either (**A**) the patches contained either buffer or *M. xanthus*, (**B**) one patch contained *E. coli* or *F. johnsoniae* and the other patch contained the same with or without pre-mixed *M. xanthus*, or (**C**) one patch contained *E. coli* or *F. johnsoniae* and the other patch contained the other prey species with or without pre-mixed *M. xanthus*. Worms found on the plate but not in a patch are indicated in yellow, and worms that had left the plate by 25 h are indicated in grey. Bars are mean worm counts from three biological replicates. Raw data are shown in Appendix A. ‘*F. john*’ = *F. johnsoniae*, *‘M.xan*_N_’, ‘*M.xan*_S_’ = *M. xanthus* strains N and S, respectively.

**Figure 4 microorganisms-09-01362-f004:**
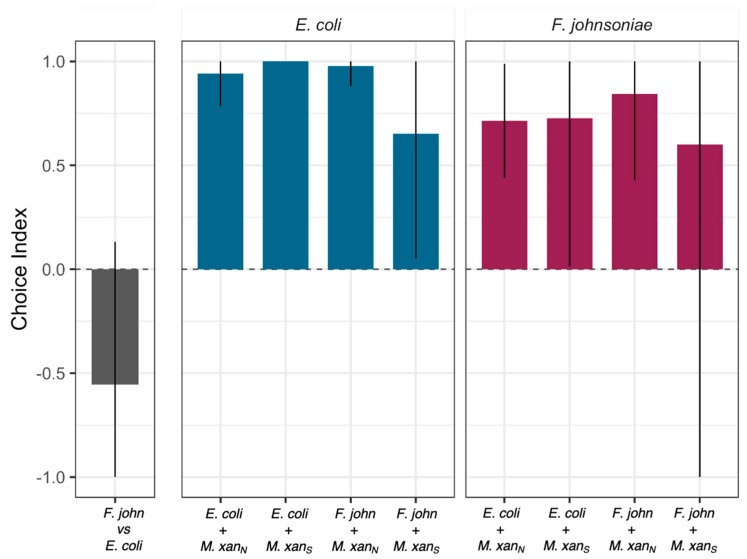
**Choice indices for circular patches of basal prey.** The leftmost section shows the prefe-rence of *C. elegans* for *E. coli* over *F. johnsoniae*. The next two sections show the choice of the worms between a mono-species patch of basal prey (indicated above the panel, *E. coli* = blue or *F. johnsoniae* = red) and a second patch (indicated on the *x*-axis) consisting of basal prey mixed with *M. xanthus* either strain S or strain N. Bars are means from three biological replicates and error bars are 95% confidence intervals. For the choice *E. coli/E. coli* + *M. xan*_S_ the error bar is zero. In some instances, 95% confidence intervals extend outside the range of what is biologically possible, [−1,1], so we restricted them to reflect the biological reality. ‘*F. john*’ = *F. johnsoniae*, ‘*M.xan_N_*’, ‘*M.xan_S_*’ = *M. xanthus* strains N and S, respectively.

**Figure 5 microorganisms-09-01362-f005:**
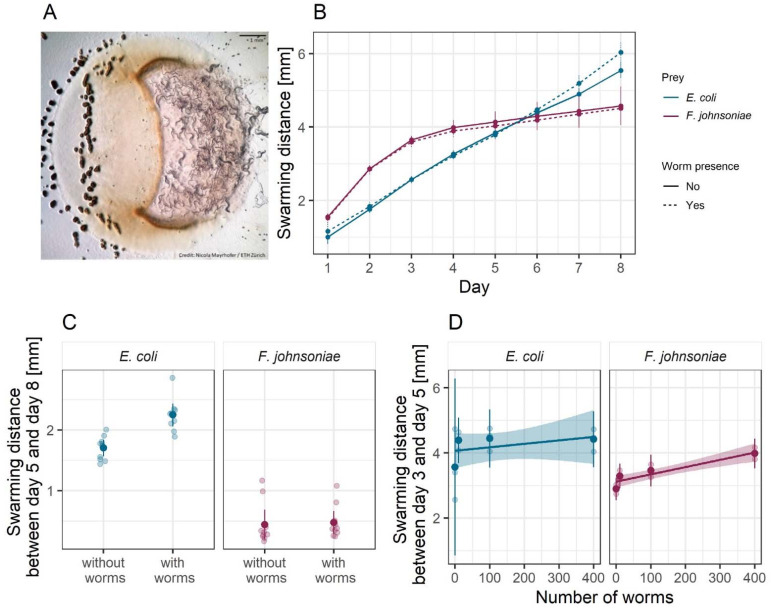
***C. elegans* presence increases *M. xanthus* predatory swarming rate**. (**A**) Picture of *C. elegans* (right) and *M. xanthus* strain S (orange lawn and black fruiting bodies on the left) preying upon an *E. coli* patch (raised circle). We estimated the predatory performance of *M. xanthus* as the swarming distance along the horizontal midline of the prey patches of *E. coli* (blue) and *F. johnsoniae* (red). We show the swarming distance over time (**B**) in the presence (dotted lines) and absence (solid lines) of *C. elegans’* populations initiated with 10 worms (which started reproducing at day 5), and the associated total swarming distances between days 5 and 8 (**C**) from the same experiment. Panel (**D**) depicts the total swarming distances between days 3 and 5 in the presence of different numbers of *C. elegans* (the worms did not reach maturity until day 5 and so did not reproduce during this experiment). Each large dot is the mean of ten (**B**,**C**) or three (**D**) biological replicates (shown as transparent dots). Error bars and shaded areas represent 95% confidence intervals of the means and the regression lines, respectively.

## Data Availability

The data presented in this study are openly available on the Dryad repository at https://doi.org/10.5061/dryad.cvdncjt3v, accessed on 15 June 2021.

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
