# Peer review of "Behavioral Interactions between Bacterivorous Nematodes and Predatory Bacteria in a Synthetic Community"

_microorganisms, 2021, doi:10.3390/microorganisms9071362_

Round 1
Reviewer 1 Report
Authors should show the growth curves.
Introduction should be more advanced (see
Biofouling 2019 Mar;35(3):273-283 ) What about the experiemntal error in Fig.2? In Fig. 3 the error is too big to make any conclusions!Author Response
Please see the attachment.

Reviewer 2 Report
This manuscript describes an experimental, synthetic food chain comprising two bacteriovorous organisms, the nematode C. elegans and the bacterium M. xanthus, which is also a potential food source for the former, and two basal prey bacteria, E. coli and F. johnsoniae. The system was designed to offer an experimental system to test ecological theories on apex predator influence on the ecology of organisms at lower trophic levels, including the effects of competition between a predicted apex predator (C. elegans) and a predicted mesopredator (M. xanthus) on predatory behaviors. The authors used choice assays with different combinations of the selected organisms to establish each member’s place on the food chain and assess how other members influence that placement. The concept of C. elegans as an apex predator of the community was not supported, since the majority of animals were repelled by living or dead M. xanthus even when it was the only food source. The authors offer various possibilities to explain this result, including that the M. xanthus secondary metabolome includes toxic or repellant compounds. Curiously the minority 10% of animals that remained preferred M. xanthus over empty agar and the authors offer the intriguing possibility that there is population heterogeneity in nematode feeding behaviors. The presence of M. xanthus can reduce the attractiveness of the basal prey. In turn, M. xanthus predation on E. coli, and to a lesser extent F. johnsoniae was enhanced by the presence of C. elegans, though the underlying cause of this observation was not explored.
Major comments:
The authors should describe their definition of mesopredator, since this classification can be murky. Further, the authors might clarify how their finding that C. elegans does not prefer to eat M. xanthus (and may avoid it) influences the potential classification of M. xanthus as a mesopredator. Do both M. xanthus and C. elegans become mesopredators in this model? Since bacteria that are classified as the same species can have a tremendous amount of genomic and functional heterogeneity, the authors should touch on why the strains that were tested chosen for the assays and how this choice might explain differences in results relative to prior publications.
The reader has to wait until the discussion for the explicit statement that the system is an artificial assembly intended to test predicted trophic positions of member organisms and how interactions may or may not influence that trophic position. A similarly explicit statement should be added to the introduction, so that the reader is better equipped to understand where data do not support the predictions, and where they do. Related to this, the food chain model predictions and findings would be more clearly made if the authors created a pictorial summary of the predicted positions of each organism on the food chain and how the data do or do not support that model. Such a diagram would be useful regardless for readers to easily keep track of what is being tested in each experiment.
The authors suggest that the influence of C. elegans on M. xanthus predation is likely indirect, citing that the nematodes rarely interact with M. xanthus when other basal prey are present. However, the data presented do not exclude the possibility that at least some C. elegans carry M. xanthus with them as they seek their basal prey, especially given the observation of behavioral heterogeneity and the relatively subtle differences in M. xanthus predation in the presence or absence of C. elegans. Even rare interactions between C. elegans and M. xanthus could contribute to this difference. The authors state that the M. xanthus swarmed faster on E. coli in the presence of C. elegans (regardless of nematode density). However, the difference in swarming depending on C. elegans presence or absence is seen only very late in the experiment (days 6-8) and at this stage of the experiment the differences could be entirely attributed to C. elegans moving M. xanthus to new locations. This concern could be addressed using C. elegans unc worms that do not move as readily. If not addressed experimentally, the authors should consider alternative explanations for these data in their discussion.
For each experiment it would help the reader tremendously to have a schematic diagram of the inoculation arrangement and capability and directionality of movement where appropriate.
In the first line of the discussion the authors suggest that there has been little research on microbial trophic chains. This may be true, but there are some studies that should be cited and discussed in the context of the authors findings. These include:
- Steffan SA, Chikaraishi Y, Currie CR, Horn H, Gaines-Day HR, Pauli JN, Zalapa JE, Ohkouchi N. Microbes are trophic analogs of animals. Proc Natl Acad Sci U S A. 2015 Dec 8;112(49):15119-24. doi: 10.1073/pnas.1508782112. Epub 2015 Nov 23. PMID: 26598691; PMCID: PMC4679051.
- Petters S, Groß V, Söllinger A, Pichler M, Reinhard A, Bengtsson MM, Urich T. The soil microbial food web revisited: Predatory myxobacteria as keystone taxa? ISME J. 2021 Mar 21. doi: 10.1038/s41396-021-00958-2. Epub ahead of print. PMID: 33746204.
Although identifying mechanistic details underlying the observations is beyond the scope and goal of the authors, the manuscript would still benefit from highlighting in the discussion the possible mechanisms that may be at work. For instance, published literature is rich with information about C. elegans avoidance behaviors and M. xanthus secondary metabolism. The authors should consider reviewing that literature to suggest known M. xanthus compounds that may already be recognized avoidance signals. In addition, the authors should consider discussing how mixtures of bacteria can produce distinctive metabolites relative to single cultures. For example:
- Stasulli NM, Shank EA. 2016 Profiling the metabolic signals involved in chemical communication between microbes using imaging mass spectrometry. FEMS Microbiol Rev. 40:807.
- Bae M, Mevers E, Pishchany G, Whaley SG, Rock CO, Andes DR, Currie CR, Pupo MT, Clardy J. Chemical Exchanges between Multilateral Symbionts. Org Lett. 2021 Mar 5;23(5):1648-1652. doi: 10.1021/acs.orglett.1c00068. Epub 2021 Feb 16. PMID: 33591189.
Minor comments:
In some figures (Fig. 1B and Fig. 3) the error bars extend beyond the range of the figure and indicate that some values can go beyond 1 and -1 in a choice index, which is somewhat confusing and would benefit from an explanation.
In the text, direct the readers to Fig. S1A, Table S1, and Fig. S4 (these are not currently referred to anywhere in the main text).
The authors appropriately show confidence intervals and describe robust statistical analyses. However, the choice of observation time points, and the impact of that choice on the results, were not adequately described. In the discussion, the authors should provide a brief acknowledgement that using discrete timepoints of observations (rather than using continuous observation with a camera) is a limitation of the study.
References 50 and 51 are the same (repeated).
Round 2
Reviewer 1 Report
Themanuscript can be accepted